# Iron (II) Metallo-Supramolecular Polymers Based on Thieno[3,2-*b*]thiophene for Electrochromic Applications

**DOI:** 10.3390/polym13030362

**Published:** 2021-01-23

**Authors:** Andrei Chernyshev, Udit Acharya, Jiří Pfleger, Olga Trhlíková, Jiří Zedník, Jiří Vohlídal

**Affiliations:** 1Faculty of Science, Department of Physical and Macromolecular Chemistry, Charles University, Hlavova 2030, 128 40 Prague 2, Czech Republic; andrei.chernyshev@natur.cuni.cz (A.C.); zednik@natur.cuni.cz (J.Z.); 2Institute of Macromolecular Chemistry, Czech Academy of Sciences, Heyrovského nám. 2, 162 06 Prague 6, Czech Republic; acharya@imc.cas.cz (U.A.); pfleger@imc.cas.cz (J.P.); trhlikova@imc.cas.cz (O.T.); 3Faculty of Mathematics and Physics, Charles University, 121 16 Prague 2, Czech Republic

**Keywords:** electrochromism, field responsive polymer, metallo-supramolecular polymer, self-assembly, terpyridine, unimer, gel electrolyte

## Abstract

Four new bis(*tpy*) unimers with different linkers between the thieno[3,2-*b*]thiophene-2,5-diyl central unit and terpyridine-4′-yl (*tpy*) end-groups: no linker (**Tt**), ethynediyl (**TtE**), 1,4-phenylene (**TtPh**) and 2,2′-bithophene-5,5′-diyl (**TtB**) are prepared, characterized, and assembled with Fe^2+^ ions to metallo-supramolecular polymers (Fe-MSPs). The Fe-MSP films prepared by spin-casting on Indium Tin Oxide (ITO) glass are characterized by atomic force microscope (AFM) microscopy, cyclic voltammetry, and UV/vis spectroscopy and studied for their electrochromism and effect of the unimer structure on their electrochromic performance. Of the studied MSPs, **Fe-Tt** shows the highest optical contrast as well as coloration efficiency (CE = 641 cm^2^ C^−1^) and the fastest optical response. This makes it an excellent candidate for possible use in electrochromic devices.

## 1. Introduction

Electrochromism is an electro-optic effect consisting of a reversible change in the material color or transparency in response to or a change in the applied electric field. Electrochromic (EC) materials are used in optical displays, smart windows, optical switching devices, camouflage objects, and in the thermal management of spacecrafts or in architecture [1,2]. In the last years, electrochromic devices are also gaining application in the field of flexible technologies such as robotics, flexible electronics, smart textiles, etc. [3]. Materials of various types, such as transition metal oxides [4,5], polyoxometalates [5], coordination compounds [5,6], conjugated polymers [2,7,8,9,10], and also some metallo-supramolecular polymers (MSPs) [7,11,12,13] are known to show this phenomenon.

A macromolecule of an MSP is composed of small or oligomeric molecules with chelate end-groups, referred to as unimers [11], which are reversibly linked into chains by coordination of the end-groups to metal ions (so-called ion couplers) [12,13,14]. Of particular interest are MSPs with reversible coordination linkages, which thus exhibit constitutional dynamics. They are polymeric in the solid-state but dissociated to oligomeric fragments in solutions and/or at elevated temperature, which gives them processing benefits.

The color and properties of MSP are tuned through the structure of the unimer and the selection of ionic couplers [13,14]. Conjugated MSPs containing ions that can be reversibly switched between different oxidation states are attractive as EC materials. They mostly operate on the basis of light absorption. MSPs assembled from bis(*tpy*) unimers (*tpy* stands for 2,2′:6′,2″-terpyridin-4′-yl end-group) and iron ions are known as high-contrast EC materials of this type. They change color by switching between oxidation state Fe(II) with intense absorption centered at ca 550 to 630 nm related to the Metal to Ligand Charge Transfer (MLCT), and Fe(III) without the MLCT band [13,14,15,16,17]. MSPs with cobalt(II) [14,18,19] and ruthenium(II) [14,18,20] ions show weaker EC effect. Another type of electrochromism exhibit the MSP composed of asymmetric bis(*tpy*) unimer and Fe^2+^ and Eu^3+^ ions in alternating arrangement [21,22]. The Eu^3+^ ions emit luminescence when iron ions are oxidized to Fe(III) state in which the MLCT is absent, while the luminescence is quenched if iron ions are switched to the state Fe(II) and the intense MLCT occurs.

MSPs derived from α,ω-bis(*tpy*) unimers with short oligophenylene [19] or oligothiophene [15,16] central units represent a significant portion of already reported electrochromic MSPs. Those derived from unimers with oligothiophene central units have been the subject of our studies [23,24,25,26] that we later on extended to include phosphole [27] and thieno[3,2-*b*]thiophene [28] central units. These new unimers and their MSPs exhibited redshifted absorption (narrower energy bandgap) and increased fluorescence quantum yield compared to the counterparts with central units composed exclusively of thiophene-2,5-diyl units. The effect of phosphole unit was attributed to the decreased aromaticity of phosphole ring while the effect of thieno[3,2-*b*]thiophene unit to its rigid coplanar structure allowing better conjugation compared to the thiophene rings linked by a single bond [29]. In this study, we present new MSPs derived from bis(tpy) unimers with thieno[3,2-*b*]thiophene central unit and various linkers (none, ethynediyl, 1,4-phenylene, and 2,2′-bithophene-5,5′-diyl), in particular the effect of the linker on the optical and redox properties of corresponding Fe-MSPs with the emphasis on electrochromism.

## 2. Materials and Methods

### 2.1. Materials

Tetrakis(triphenylphosphane)palladium(0), triphenylphosphane, palladium(II) acetate, 2,5-bis(trimethylstannyl)thieno[3,2-*b*]thiophene, 2,2′-bithiophene-5-boronic acid pinacol ester, iron(II) perchlorate hydrate, copper(I) iodide, propylene carbonate (all Merck, KGAA, Darmstadt, Germany), tetrabutylammonium hexafluorophosphate (≥99.0%, NBu_4_PF_6_) lithium perchlorate (both Fluka, VWR International, Stribrna Skalice. Czech Republic), poly(methyl methacrylate) (≥99%), *N*-bromosuccinimide (NBS) (98%), tetrahydrofurane (≥99.9%THF) (all Sigma Aldrich, Prague, Czech Republic), ethynyltrimethylsilane, thieno[3,2-*b*]thiophene (both ABCR, Karlsruhe, Germany), 1,1,1,3,3,3-hexafluoropropan-2-ol (HFP) (Fluorochem, Hadfield, Derbyshire, UK), piperidine (Acros Organics, Thermo-Fisher, Prague, Czech Republic), 4′-bromo-2,2′:6′,2″-terpyridine, 4′-(4-bromophenyl)-2,2′:6′,2″-terpyridine (both TCI, Tokyo, Japan), N,N-dimethylformamide and toluene (both of the HPLC grade, VWR International, Stribrna Skalice. Czech Republic), potassium carbonate (Lach-Ner, Neratovice, Czech Republic), acetonitrile (ACN) (Uvasol, Germany, VWR International, Stribrna Skalice, Czech Republic) and Indium Tin Oxide (ITO) (20 Ω/square, Ossila Ltd., Sheffield, UK) were used as received.

### 2.2. Methods

NMR spectra were recorded on a Bruker Avance III 600 MHz instrument in 1,1,2,2-tetrachloroethane-*d*_2_ (TCE-*d*_2_) at 110 °C, referenced to the solvent signals (6.0 ppm for ^1^H and 73.8 ppm for ^13^C) and deciphered using the first-order analysis. UV/vis spectra were recorded on a Shimadzu UV-2401PC instrument (Shimadzu, Prague, Czech Republic) at room temperature using quartz cuvettes of a 0.4 cm optical path. IR spectra were recorded on a Thermo Nicolet 7600 FTIR spectrometer equipped with a Spectra Tech InspectIR Plus microscopic accessory (Thermo-Fisher, Prague, Czech Republic) using KBr-diluted samples and the diffuse reflectance technique (DRIFT). Atomic absorption spectra (AAS) of ACN solutions of Fe(ClO_4_)_2_ hydrate were measured on a Perkin Elmer AAS spectrometer (model 3110) (Rodgau, Germany) using a hollow cathode lamp emitting a spectrum specific to Fe and a commercial standard (Analytika spol. s.r.o., Prague, Czech Republic) for external calibration.

MALDI-TOF mass spectra were acquired using an UltrafleXtreme (Bruker Daltonics, Bremen, Germany) in the positive ion reflectron mode (sum of 25,000 shots with a DPSS Nd: YAG laser (355 nm, 2000 Hz) and external calibration for molecular weight assignment. About 1 mg of ground sample was deposited on the ground-steel target plate, pressed with a spatula, overlaid by 1 µL matrix solution of trans-2-[3-(4-*tert*-butylphenyl)-2-methyl-2-propenylidene]malonitrile (≥98.0%, Sigma–Aldrich, Prague, Czech Republic) in THF (10 mg mL^−1^) and dried at ambient atmosphere.

The topography of Fe^2+^-MSP films deposited by spin coating on indium tin oxide coated glass substrates was studied in the air using a Nanoscope IIIa (Veeco Instruments New York, NY, USA)) atomic force microscope (AFM) operating in the tapping mode (OTESPA-R3, Bruker silicon tips with spring constant of k = 26 N/m and resonance frequency of 300 kHz). The scans of 2 × 2 µm were acquired with scan rates of 0.7 Hz. The NanoScope Analysis software was used to process raw images and to estimate the root mean squared roughness. AFM was used to measure the thickness and surface roughness of spin-cast films.

Spectro-electrochemical and kinetic studies were performed using UV-Vis-near-infrared (NIR) Lambda 950 spectrometer (Perkin Elmer, Beaconsfield, UK) in combination with Bio-logic potentiostat/galvanostat VSP300. Three-electrode setup was used: spin-cast Fe-MSPs on fluorine doped ITO glass as working electrode, and a platinum wire and Ag/Ag^+^ electrodes were used as counter and pseudo-reference electrodes. The measurements were performed in an ACN solution of NBu_4_PF_6_ (0.1 M) in a quartz cuvette (1 cm) at room temperature. Potentials versus the Ag/Ag^+^ pseudo-reference electrode are reported.

### 2.3. Synthesis of Unimers

Prepared unimers are depicted in Scheme 1 including numbering related to the NMR assignment.


*General Procedure for the Stille Coupling Reactions*


A measured amount (2 equivalents, eq) of a given bromo-derivative was added to a solution of 2,5-bis(trimethylstannyl)thieno[3,2-*b*]thiophene (1 eq) in DMF (7 mL) and the resulting solution was bubbled with argon for 15 min. Then Pd(PPh_3_)_4_ (0.1 eq) was added under argon stream and the mixture heated and kept under stirring at 110 °C overnight. After that the reaction mixture was cooled in a freezer and the obtained sediment was filtered, washed with distilled water (3 × 25 mL), *n*-hexane (3 × 25 mL), toluene (3 × 25 mL) and finally dried for a few days to obtain the desired product as a colored powder in the isolated yield of 72 to 82%.


***Tt***
*—2,5-bis(2,2′:6′,2″-terpyridine-4′-yl)thieno[3,2-b]thiophene*


Green powder (211 mg, 82%).

IR (DRIFT): see Appendix A in ESI.

^1^H NMR (600 MHz, TCE-*d*_2_, 110 °C) δ 8.84–8.79 (m, 8H, H^5^+H^8^), 8.68 (d, *J* = 7.9 Hz, 4H, H^11^), 8.03 (s, 2H, H^2^), 7.92 (td, *J*_1_ = 7.7 Hz, *J*_2_ = 1.7 Hz, 4H, H^9^), 7.42–7.38 (m, 4H, H^10^) (see Appendix A in Electronic Supporting Information, ESI).

^13^C NMR (151 MHz, TCE-*d*_2_, 110 °C) δ 156.4 (4C, C^6^), 156.0 (4C, C^7^), 149.1 (4C, C^11^), 144.9 (2C, C^1^), 143.3 (2C, C^3^), 140.7 (2C, C^4^), 136.4 (4C, C^9^), 123.6 (4C, C^10^), 121.1 (4C, C^8^), 118.1 (2C, C^2^), 117.2 (4C, C^5^) (see Appendix A in ESI).

MALDI-TOF MS found *m*/*z*: 603.17; theory for C_36_H_22_N_6_S_2_ [M + H]^+^: 603.13.


***TtPh***
*—2,5-bis{4-(2,2′:6′,2″-terpyridine-4′-yl)-phenyl}thieno[3,2-b]thiophene*


Orange powder (185 mg, 76%).

IR (DRIFT: see Appendix A in ESI.

^1^H NMR (600 MHz, TCE-*d*_2_, 110 °C) δ 8.84 (s, 4H, H^9^), 8.80 (d, *J* = 3.9 Hz, 4H, H^12^), 8.71 (d, *J* = 7.9 Hz, 4H, H^15^), 8.01 (d, *J* = 8.2 Hz, 4H, H^5^), 7.92 (td, *J*_1_ = 7.7 Hz, *J*_2_ = 1.6 Hz, 4H, H^13^), 7.86 (d, *J* = 8.2 Hz, 4H, H^6^), 7.65 (s, 2H, H^2^), 7.41–7.38 (m, 4H, H^14^) (see Appendix A, ESI).

^13^C NMR (151 MHz, TCE-*d*_2_, 110 °C) δ 156.3 (4C, C^10^), 156.2 (4C, C^11^), 149.2 (2C, C^8^), 149.0 (4C, C^15^), 145.5 (2C, C^3^), 139.9 (2C, C^1^), 138.0 (2C, C^7^), 136.4 (4C, C^13^), 135.3 (2C, C^4^), 127.8 (4C, C^6^), 126.2 (4C, C^5^), 123.4 (4C, C^14^), 121.1 (4C, C^9^), 118.6 (4C, C^12^), 116.0 (2C, C^2^) (see Appendix A, ESI).

MALDI-TOF MS found *m*/*z*: 755.20; theory for C_48_H_30_N_6_S_2_ [M + H]^+^: 755.20.


***TtB***
*—2,5-bis{5-(2,2′:6′,2″-terpyridine-4′-yl)-2,2′-bithiophen-5′-yl}thieno[3,2-b]thiophene*


Dark-red powder (116 mg, 72%).

IR (DRIFT): see Appendix A in ESI.

^1^H NMR (600 MHz, TCE-*d*_2_, 110 °C) δ 8.81 (ddd, *J*_1_ = 4.7 Hz, *J*_2_ = 1.6 Hz, *J*_3_ = 0.8 Hz, 4H, H^16^), 8.75 (s, 4H, H^13^), 8.68 (d, *J* = 7.9 Hz, 4H, H^19^), 8.02 (s, 2H, H^2^), 7.93 (td, *J*_1_ = 7.7 Hz, *J*_2_ = 1.7 Hz, 4H, H^17^), 7.75 (d, *J* = 3.8 Hz, 2H, H^10^), 7.42-7.40 (m, 4H, H^18^), 7.33 (d, *J* = 3.8 Hz, 2H, H^9^), 7.29 (d, *J* = 3.8 Hz, 2H, H^5^), 7.25 (d, *J* = 3.8 Hz, 2H, H^6^) (see Appendix A in ESI).

^13^C NMR (151 MHz, TCE-*d*_2_, 110 °C) δ 156.2 (4C, C^14^), 156.0 (4C, C^15^), 148.9 (4C, C^19^), 142.8 (2C, C^1^), 141.0 (2C, C^12^), 139.0 (2C, C^3^), 138.7 (2C, C^11^), 138.4 (2C, C^7^), 137.0 (2C, C^8^), 136.5 (4C, C^17^), 136.4 (2C, C^4^), 126.6 (2C, C^2^), 125.0 (2C, C^9^), 124.8 (2C, C^6^), 124.8 (2C, C^5^), 123.6 (4C, C^18^), 121.2 (4C, C^16^), 117.0 (4C, C^13^), 115.9 (2C, C^10^) (see Appendix A in ESI).

MALDI-TOF MS found *m*/*z*: 931.05; theory for C_52_H_30_N_6_S_6_ [M + H]^+^: 931.09.


***TtE***
*—2,5-bis{1-(2,2′:6′,2″-terpyridine-4′-yl)-ethyn-2-yl}thieno[3,2-b]thiophene*


Red powder (198 mg, 38%).

**TtE** was prepared using Sonogashira coupling. 4′-Bromo-2,2′:6′,2″-terpyridine (497 mg, 1.59 mmol) was added to a solution of 2,5-bis(ethynyl)thieno[3,2-*b*]thiophene (150 mg, 0.79 mmol) in piperidine (10 mL), the obtained solution was bubbled with argon for 10 min and Pd(PPh_3_)_4_ (92.0 mg, 0.079 mmol) and CuI (15.0 mg, 0.079 mmol) were added. Then the reaction mixture was again bubbled with argon for 10 min, heated, and kept at 100 °C overnight. After that the reaction mixture was cooled in a freezer, the obtained sediment was filtered, washed with distilled water (3 × 25 mL), *n*-hexane (3 × 25 mL), toluene (3 × 25 mL) and finally dried for a few days to obtain the desired product as a red powder in the isolated yield of 198 mg (38%).

IR (DRIFT): see Appendix A in ESI.

^1^H NMR (600 MHz, TCE-*d*_2_, 110 °C) δ 8.79 (d, *J* = 4.0 Hz, 4H, H^10^), 8.67–8.63 (m, 8H, H^7^+H^13^), 7.92 (td, *J*_1_ = 7.7 Hz, *J*_2_ = 1.7 Hz, 4H, H^11^), 7.58 (s, 2H, H^2^), 7.40 (ddd, *J*_1_ = 7.4 Hz, *J*_2_ = 4.7 Hz, *J*_3_ = 1.0 Hz, 4H, H^12^) (see Appendix A in ESI).

^13^C NMR (151 MHz, TCE-*d*_2_, 110 °C) δ 155.7 (4C, C^8^), 155.6 (4C, C^9^), 149.1 (4C, C^13^), 140.1 (2C, C^1^), 136.6 (4C, C^11^), 132.3 (2C, C^6^), 126.3 (2C, C^3^), 125.0 (2C, C^2^), 123.7 (4C, C^7^), 122.4 (4C, C^12^), 121.1 (4C, C^10^), 94.0 (2C, C^5^), 87.1 (2C, C^4^) (see Appendix A in ESI).

MALDI-TOF MS found *m*/*z*: 651.17; theory for C_40_H_22_N_6_S_2_ [M + H]^+^: 651.13.

## 3. Results and Discussion

### 3.1. Synthesis of Unimers

Synthetic pathways to prepared unimers are depicted in Scheme 2. Unimers **Tt, TtPh,** and **TtB** were prepared from commercially available 2,5-bis(trimethylstannyl)-thieno[3,2-*b*]thiophene by using Stille coupling. Commercially available bromo derivatives were used for the syntheses of Tt and TtPh unimers while the bromo derivative necessary for preparing **TtB** unimer [4′-(5-bromo-2,2′-bithiophen-5′-yl)-2,2′:6′,2″-terpyridine] was prepared according to the procedure described earlier [24]. Unimer **TtE** was prepared using the Sonogashira coupling approach. The necessary precursor: 2,5-bis(ethynyl)thieno[3,2-*b*]thiophene was prepared according to the procedure described in [30] and used immeditely after preparation, or stored in a freezer so that it does not decompose after spontaneous deprotection (which is the reason for the lower yield of unimer TtE).

The prepared unimers have been obtained as powdered solids of low solubility in common organic solvents, which made their isolation easy. Their insolubility can be attributed to π-stacking of unsubstituted aromatic subunits. 1,1,1,3,3,3-Hexafluoropropan-2-ol (HFP) and its mixtures with acetonitrile (ACN) were found as good solvents of unimers, as well as related MSPs at room temperature, which might be due to acidity of HFP [31] providing effective reversible interactions with nitrogen atoms of *tpy* groups. As indicated in the Experimental section, unimers have been characterized by the MALDI-TOF MS, IR, ^1^H and ^13^C NMR (Appendix A in ESI), and UV/vis spectroscopies.

### 3.2. Self-Assembly of Fe-MSPs from Unimers and Fe^2+^ Ions in HFP/ACN Solutions

Thermodynamically driven assembly of Fe-MSPs (Scheme 3) in HFP/ACN mixed solvent (4/1 by vol.) was monitored by UV/vis spectroscopy at room temperature using the following procedure.

A series of 13 solutions of practically the same unimer concentration (2 × 10^−5^ M) and the Fe^2+^ ions-to-unimer mole ratio *r =* [Fe^2+^]/[U] increasing stepwise from 0 to 3 was prepared for each unimer by mixing 2 mL of the unimer stock solution (2 × 10^−5^ M) in the HFP/ACN mixed solvent with a calculated volume (from 2 to 60 μL) of an Fe(ClO_4_)_2_ solution in ACN (2 × 10^−3^ M). Dilution caused by the added Fe(ClO_4_)_2_ solution was up to 3%, which can be considered negligible. Since Fe(ClO_4_)_2_ hydrate is hygroscopic and therefore contains an undefined amount of water, the exact concentration of Fe^2+^ in its stock ACN solution was determined by atomic absorption spectrometry (AAS). Preparation of the unimer stock solutions was finalized by ultrasonication, immersing a flask with the solution into an ultrasound bath for 5 min. The prepared solutions were allowed to equilibrate for at least half a day before their spectra were measured.

The UV/vis spectra of prepared solutions of different mole ratios *r* (i.e., of different compositions) are shown in Figure 1 together with photos of vials filled with the respective solutions. Changes in color are visible by the naked eye and they clearly show the great effect of linkers connecting chelate groups to the central unit on the optical spectrum of particular MSPs (see also Appendix A). The observed red color of **Fe-TtPh** is quite atypical because the Fe^2+^(*tpy*)_2_ complexes with an MLCT band are usually blue or green [12,32]. The UV/vis spectrum indicates that the red color stems from the relatively high intensity of the bands contributed with electronic transitions in *tpy* end-groups (around 280 nm) and in the linkers and central unit unimeric units (ranging from 300 to ca 500 nm) and the relatively low intensity of the MLCT band (see the comparison in Appendix A). Of course, this effect might be used for fine color tuning of similar MSP systems.

The spectral changes in response to the increasing *r* ratio show three stages. The first stage for *r* from 0 to 0.5 is characterized by the redshift of the band unimeric units (U band) with a maximum at the wavelength *λ*_U_ in the range from 400 to 500 nm, and by appearance and enhancement of the MLCT band, whose apex wavelength, *λ*_MLCT_, is a function of the linker structure (see Figure 2 and Appendix A). These features correspond to the formation of the species U-Fe^2+^-U, called ‘butterfly dimers’ [32]. The observed slight redshift of the MLCT bands in this stage is caused by the broadening and redshift of a particular U-band.

The second stage occurs for *r* from 0.5 to 1.0, and is characterized by a further slight redshift of *λ*_U_ as well as *λ*_MLCT_ and significant enhancement of the MLCT band. These features prove the assembling of butterfly dimers and Fe^2+^ ions into polymeric chains, whose degree of polymerization should be the highest for the composition ratio *r* equal to approximately one. The total redshift *λ*_U_ from unimer (*r* = 0) to the ideal MSP (*r* = 1) is about 20 nm for all MSPs (see Table 1).

The third stage occurs for *r* values above 1 and is usually characterized by a blue shift and attenuation of the MLCT band with increasing *r*, which mostly accompanies partial dissociation of longer MSP chains to the shorter ones end-capped with Fe^2+^ ions [24,25,26,33]. However, **Fe-TtB** showed the great exception: continuing redshift and amplification of the MLCT band, which is most likely due to the large overlap of the MLCT and U bands in this MSP. It is worth noting that according to the optical absorption spectrum this unimer clearly shows the highest extent of electron delocalization among all unimers in this study.

### 3.3. Preparation and Characterization of Fe-MSP Films

The films were prepared by casting from Fe-MSP solutions (2.6 g/L) in HFP/ACN mixed solvent, which were prepared by mixing the solutions containing equimolar amounts of the components. The solutions were allowed to equilibrate until the next day. Prior to the film casting, a particular solution was treated in an ultrasound bath for 5 min and filtered. Then it was spin-casted (500 rpm, 30 s) onto a rectangular ITO substrate and dried in air. The morphology, roughness, and thickness of spin-cast Fe-MSP films were analyzed by the AFM method. A relatively rough surface was observed for all films (see Appendix A in ESI). The surface roughness was found to increase with the increased length of the linker of the respective unimer. The smoothest (least coarse) film showed **Fe-Tt** with the difference between the highest and the lowest point of the surface equal to 23 nm. The average film thickness was determined to vary from 35 to 40 nm (see Table 2).

The redox behavior of Fe-MSP films was determined by cyclic voltammetry using an oxygen-free solution of NBu_4_PF_6_ (0.1 M) in ACN as the electrolyte and the scan rate ranged from 20 to 1000 mV/s (see Figure 3). A single reversible redox wave corresponding to the transformation between Fe^2+^ and Fe^3+^ ions has been observed for all MSPs, in good agreement with the behavior of related systems [8]. The observation that the separation between the anodic and cathodic peaks increases with the increasing scanning speed suggests that the redox reaction is quasi-reversible [11,12]. The insets in Figure 3 show that the peak current density is directly proportional to the square root of the scan speed for all MSPs, which indicates a diffusion control of the redox process in all systems regardless of the linker structure. The noticeable smaller decrease in the proportionality constant of this dependence with increasing linker size can be attributed to the larger volume of unimer units and thus to the lower concentration of Fe ions in the corresponding systems.

### 3.4. Electrochromism of Prepared MSPs

The UV/vis to near-infrared (NIR) absorption spectra of Fe-MSP films as well as photos of the films taken prior and after oxidation of Fe^2+^ ions are shown in Figure 4. Relating spectral data are summarized in Table 2. It can be seen that, after the electrochemical oxidation of Fe^2+^ to Fe^3+^, the MLCT band almost disappeared and the unimer band was red-shifted significantly. The color of the oxidized film is dominantly controlled by the U band, which, although shifted, remained clearly distinguished and intense only in Fe-MSPs with no or a short linker. In the oxidized Fe-MSPs with longer conjugated linkers, the U band is strongly broadened. In the Fe^3+^-TtB spectrum, this band is almost invisible. Another interesting feature of the Fe^3+^-TtB film is an intense absorption band in the NIR region with a maximum at about 1045 nm.

The MLCT band is almost fully reversible, giving upon reduction the film in its original color. The results of responsivity tests during periodic cycling of the applied voltage between 0 and 1.4 V are shown in Figure 5 and the results of chronoamperometry measurements in Figure 6.

The optical contrast (Δ*T*) measured at corresponding wavelengths (Table 3) was found to be 29.0% for **Fe-TtB** and increased to 64.5% for **Fe-Tt**. For comparison, Zhu et al. reported for electrochromic conjugated polymers based on thieno[3,2-*b*]thiophene the optical contrast values 19.15% and 13.36% in a similar spectral region [34]. The film **Fe-TtB** also showed optical contrast at 1041 nm with Δ*T* = 48% and Δ*OD* = 0.35, but it disappeared after few repetition cycles. Bleaching and coloring times reported in Table 3 are the times required for reaching 95% of the final transmittance at given wavelengths after switching the applied voltage (see Appendix A in ESI). The coloring time, *t*_c_, (reduction process) was nearly the same (1.7 to 1.8 s) for all Fe-MSPs, while the bleaching time, *t*_b_, (oxidation process) differed up to more than fourfold. The linear dependences shown in the insets of Figure 3 correspond to the Randles-Sevcik equation [35,36] (see page 9 in ESI). Slopes of these dependences decreased from the Fe-Tt to Fe-TtB with increasing size of the parent unimer (with only one exception, see Table 3). This decrease can therefore be tentatively attributed to the gradually decreasing concentrations of the active redox species in the EC material. However, these dependencies do not provide an explanation for the substantially prolonged bleaching times of Fe-TtE and Fe-TtB. Furthermore, there is no correlation between bleaching time and surface roughness of the EC layer. Thus, the differences in bleaching time between different Fe-MSPs are most likely due to differences in the rate of diffusion of counter-ions into polymer films during their oxidation and the ion/charge transport between EC film and electrolyte [37]. The reverse diffusion of counter-ions out of partly swollen film during the reduction process should be easier, which might explain the observed similarity of *t*_c_ values.

One of the most important parameters for the characterization of EC materials is coloration efficiency, *CE* = Δ*OD/Q*, defined as the difference in optical density Δ*OD* = log (*T*_b_/*T*_c_) per unit charge density *Q* measured at the *λ*_max_ of the optical absorption band [38,39,40]. We adopted the approach suitable for not fully bleached EC materials [41] and used the Q values needed to achieve a 95% final transmittance change to calculate the CE values (see Appendix A). The remaining 5% is barely visible to the naked eye. The *Q* values were calculated directly by the instrument used for chrono-coulometric measurements as the time integral of the current across the corresponding interval.

The highest coloration efficiency, 641 cm^2^ C^−1^, was found for **Fe-Tt** film. This value is very competitive compared to other materials reported in the literature for electrochromic devices. For example, Mondal [40] and Kuai [41] reported CE values between 12.84 and 230 cm^2^ C^−1^ for their Fe(II) coordination nanosheets, Kuo et al. [38] achieved CE = 372.7 cm^2^ C^−1^ for copolymer composed of carbazole and indole-6-carboxylic acid.

### 3.5. Fabrication of Electrochromic Devices

The solid-state electrochromic device has been designed, constructed, and its operation has been proven (see Figure 7). Here we present the device based on **Fe-Tt,** the MSP with the fastest EC response (Table 3) and the highest change in transmittance. An **Fe-Tt** film spin-casted on ITO glass was covered with the gel electrolyte layer (thickness of 1 mm) prepared by mixing poly(methyl methacrylate) (3.5 g), propylene carbonate (10 mL), and LiClO_4_ (1.5 g) according to [14], and stacked with other ITO substrates (see Figure 7). The ionic conductivity of the gel electrolyte was found to be 2.3 × 10^−3^ S/cm, in good agreement with the earlier reported values [42]. This value has been determined by the broadband impedance spectroscopy from the plateau of the frequency dependence of the real part of the conductivity above 10^3^ Hz (Appendix A) and confirmed by the Nyquist plot considering Randles circuit. The UV/vis transmittance spectra of the device (Figure 8) showed optical contrast of around 60% at 637 nm, which is lower than the value listed in Table 3 for an **Fe-Tt** film in a liquid electrolyte. Longer coloration and bleaching times can be attributed to the lowered conductivity of ions in the gel electrolyte compared to that of the liquid electrolyte. On the other hand, the gel electrolyte provides higher stability to the electrochromic cell during cycling.

The Fe-MSP films showed good stability for a few months when stored in the air under ambient conditions. However, the Fe-MSP device with liquid electrolyte showed signs of degradation after 20 working cycles, especially delamination of the EC layer and obvious signs of its partial dissolution. Degradation could be only partially limited by covering the Fe-MSP layer with Nafion. In contrast, the Fe-MSP based device with the gel electrolyte showed markedly high stability during electrochromic cycling: no change was detected during 100 cycles. We should also note that some degradation over the long-term may occur due to increasing resistance of the ITO layer during repeated redox cycling. Fluorine-doped tin oxide (FTO) would therefore be a better choice for long term working devices.

## 4. Conclusions

The results obtained clearly show how the optical spectral properties of ditopic unimers and particularly spectral and electrochromic properties of related Fe-MSPs can be simply tuned via the choice of linkers connecting chelate end-groups to the unimer central unit. The presented four new unimers of the a,w-bis(*tpy*) family with thieno[3,2-*b*]thiophene-2,5-diyl central unit differed only in the linkers (none, ethynediyl, 1,4-phenylene, and 2,2′-bithophene-5,5′-diyl). The unimers are easily assembled with Fe^2+^ ions to give electrochromic Fe-MSPs of ground colors ranging from blue-green to gray-green, red, and deep purple. The red color of Fe-MSPs is rather surprising because the “generic” color of MSPs with [Fe(tpy)_2_]^2+^ centers is blue due to the very intense MLCT band occurring at 550 nm and above [12,32]. Nevertheless, it is also well known that this MLCT band is significantly contributed by transitions in the central parts of unimeric units [24,25,26,27]. This interesting feature can potentially help in easy color tuning the EC devices based on bis(*tpy*) Fe-MSPs because their electrochromism is closely related to the color changes accompanying the MSP assembling, which are easy to determine.

One can see that the linker-less **Fe-Tt** provided the Fe-MSP with the highest optical contrast, fastest response, and the highest coloration efficiency (CE = 641 cm^2^ C^−1^) of all examined Fe-MSPs. To the best of our knowledge, its properties are comparable to the highest values reported for MSPs in the literature, which makes **Fe-Tt** an excellent candidate for possible applications in electrochromic devices. On the other hand, **Fe-TtB** possessing the largest linkers (2,2′-bithiofen-5,5′-diyl) exhibited the worst of both of these characteristics, but showed a rare color change (green to dark purple) and, in addition, showed rather high NIR electrochromism (optical contrast about 50% at 1100 cm^−1^), though only during the first few cycles.

It should be noted here that the Fe-MSP films derived from conjugated bis(*tpy*) unimers cannot be turned to a fully transparent state. Unsubstituted bis(*tpy*) complexes of metal ions absorb in the near UV region (band edge around 350 nm) and small conjugated substituents attached to *tpy* redshift the band edge of the complex to the UV-visible border 12,32]. In addition Fe ions show a weak band centered at 370 nm contributed by transitions at Fe atoms [43]. Considering conjugated Fe-MSPs, the redox process taking place in [Fe(*tpy*)_2_]^2+^ centers always affects optical transitions in unimeric units which significantly contribute to the MLCT band [24,25,26,27]. Besides, eventual simultaneous changes in populations of polaronic and bipolaronic states enable optical transitions that make a fully bleached state principally unattainable and the design of organic EC devices difficult. On the other hand, these materials require a mostly lower amount of charge compared to inorganic EC materials to achieve the same color change, because of the higher oscillator strengths of the associated electronic transitions.

## Data Availability

We have no depository of publicly archived datasets analyzed or generated during the study. Data are available on request; contact please authors on their E-mail addresses.

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
