# Peer review of "Iron (II) Metallo-Supramolecular Polymers Based on Thieno[3,2-b]thiophene for Electrochromic Applications"

_polymers, 2021, doi:10.3390/polym13030362_

Round 1

Reviewer 1 Report

The paper is well and carrefully written. The authors shoul move the materials part before the methods and pay attention with the abreviation os the compounds. For instance should define ACN (line 72), THF (line 79), NBu4PF6 (line 92), DMF (line 103). At the same tome they should provide the purity grade of all the reagents and include in this section PMA, propylene carbonate, LiCLO4, ITO.

Why do not present the IR spectra, even in suplementary information, when they present the NMR spectra?

How can the authors determine the Q fromthe CA measurements? Can they give at  least one example?

The section 3.5 should be improved, with a better description of the experiment and the presentation of the main results.

Reviewer 2 Report

The manuscript "Iron (II) metallo-supramolecular polymers based on 2 thieno[3,2-b]thiophene for electrochromic applications" presents very exciting results that without question are of great interest in the electrochromic community. The authors present an interesting approach to obtain distinct colors utilizing different linkers between thieno[3,2-b]thiophene (TT) and tpy chelating end-groups on the electrochromic properties of Fe-MSP molecules. This work is especially useful for color tuning based on TT derivatives. Experimental results and imterpretations are sound. And thus I recommend publishing this manuscript. The specific comments are as follows:

1) What is the ionic conductivity and thickness of the electrolytes gel used in the sandwich-type device (Fig 7)? I suggest authors to include this value in main text or ESI.

2) Please comment on the delamination and/or mechanical stability of electrochromic layers in the presence of the electrolytes.

3) One of the merits of a organic/polymer ECD is to obtain the flexibility of the device. It should be good idea if the merits are addressed in the introduction section.

4) Please comment on the long-term stability (switchability) of the Fe-MSP based ECDs and what would be the possible degradation mechanism in this case.

5) I commend authors for including switching time studies (Figure S12). I would suggest authors to include a brief discussion on why Fe-TtE and Fe-TtB have a slower bleaching time (tb) compared to Fe-Tt and FeTtPh. You can discuss in terms of ion/charge transport between EC film and electrolyte. I included a paper below you can reference on

Zhu, et al. "Electrochromic properties as a function of electrolyte on the performance of electrochromic devices consisting of a single-layer polymer." Organic Electronics 15.7 (2014): 1378-1386. https://doi.org/10.1016/j.orgel.2014.03.038

6) The references in this manuscripts regarding thieno[3,2-b]thiophenes (TT) are few. Below are some latest works focus on the use of TT for electrochromic devices. You can use them in conjunction with ref 31 or introduction section.

Li, et al. "Solution-processable neutral green electrochromic polymer containing thieno [3, 2-b] thiophene derivative as unconventional donor units." Macromolecules 49.19 (2016): 7211-7219. https://pubs.acs.org/doi/10.1021/acs.macromol.6b01624

Otley, et al. "Color‐Tuning Neutrality for Flexible Electrochromics Via a Single‐Layer Dual Conjugated Polymer Approach." Advanced Materials 26.47 (2014): 8004-8009. https://doi.org/10.1002/adma.201403370

7) One of my main concerns about the presented technology is the lack of a transparent to colored transition. While multicolor transitions may be appealing, they are difficult to use in real devices. For instance, if the authors would like to develop a full color pixel based on a monolithic architecture, they will not have the possibility of showing a non-colored state, which greatly lowers the future development of this technology, and restricts it to a limited number of applications. The authors may have these aspects in mind in order to try to make a broader discussion on the future of this technology.

Round 2

Reviewer 1 Report

I am pleased with the authors answers.